# The Influence of an Alpha Band Neurofeedback Training in Heart Rate Variability in Athletes

**DOI:** 10.3390/ijerph182312579

**Published:** 2021-11-29

**Authors:** Christophe Domingos, Carlos Marques da Silva, André Antunes, Pedro Prazeres, Inês Esteves, Agostinho C. Rosa

**Affiliations:** 1Life Quality Research Centre, 2040-413 Rio Maior, Portugal; csilva@esdrm.ipsantarem.pt; 2Laboratory of Physiology and Biochemistry of Exercise, Faculty of Human Kinetics, University of Lisbon, 1495-751 Oeiras, Portugal; a.s.t.antunes.7@gmail.com; 3Faculty of Health Sciences and Sport, University of Stirling, Stirling FK9 4LJ, UK; pedronprazeres@gmail.com; 4Department of Bioengineering, LaSEEB—System and Robotics Institute, Instituto Superior Técnico, University of Lisbon, 2695-066 Lisbon, Portugal; ines.s.esteves@gmail.com (I.E.); acrosa@laseeb.org (A.C.R.)

**Keywords:** electroencephalography, neurofeedback, biofeedback training and RMSSD

## Abstract

Neurofeedback training is a technique which has seen a widespread use in clinical applications, but has only given its first steps in the sport environment. Therefore, there is still little information about the effects that this technique might have on parameters, which are relevant for athletes’ health and performance, such as heart rate variability, which has been linked to physiological recovery. In the sport domain, no studies have tried to understand the effects of neurofeedback training on heart rate variability, even though some studies have compared the effects of doing neurofeedback or heart rate biofeedback training on performance. The main goal of the present study was to understand if alpha-band neurofeedback training could lead to increases in heart rate variability. 30 male student-athletes, divided into two groups, (21.2 ± 2.62 year 2/week protocol and 22.6 ± 1.1 year 3/week protocol) participated in the study, of which three subjects were excluded. Both groups performed a pre-test, a trial session and 12 neurofeedback sessions, which consisted of 25 trials of 60 s of a neurofeedback task, with 5 s rest in-between trials. The total neurofeedback session time for each subject was 300 min in both groups. Throughout the experiment, electroencephalography and heart rate variability signals were recorded. Only the three sessions/week group revealed significant improvements in mean heart rate variability at the end of the 12 neurofeedback sessions (*p* = 0.05); however, significant interaction was not found when compared with both groups. It is possible to conclude that neurofeedback training of individual alpha band may induce changes in heart rate variability in physically active athletes.

## 1. Introduction

Heart rate variability (HRV) is a measure of interest in the sports domain since it has been linked with autonomic nervous system (ANS) function and cardiovascular control. This system has been difficult to train/influence due to is nature in controlling the involuntary functions related to the organism [1]. It can be divided into two subsystems: sympathetic and parasympathetic. A greater activation of the former is observed in stressful situations, while the latter is more active in the recovery phases [2].

It has been reported that a decreased HRV is associated with worse clinical states [3,4], such as cardiovascular pathologies [5], panic disorder [6], depression [7] and anxiety [8] while an increased HRV has been positively associated with physiological health status [9], but only if a parasympathetic activity augmentation occurs and not a sympathetic activity augmentation [10,11]. Furthermore, relations have been established between HRV metrics and cognitive performance and emotional regulation [12,13], in which high values correspond to an increased ability of information processing and attentional focus maintenance, suggested as important aspects to respond to changes in environment conditions [14,15]. Conversely, a lower HRV seems to be related to a greater sympathetic hyper-reactivity [16], which in turn is related to high stress [17], which has been appointed as a factor that can negatively impact sports performance [18].

From a technical perspective, HRV corresponds to the observed variation of the time interval between heart beats—RR intervals in the electrocardiogram signal [19], relevant in studying the cardiac behaviour in different contexts [20] and in gauging the state of the ANS [21]. The HRV temporal signal can be decomposed in a frequency power spectrum [22], with high frequency bands indicating prevalence of parasympathetic activity in the modulation of vegetative and cardiac activity. For short duration collections (≤ 5 min), the variable showing more consistency analysis is the root mean square of successive difference of the R–R intervals (RMSSD), which has been shown to have higher reproducibility [23].

An increase in psychological stress levels causes a decrease in the intervals measured in the high frequency bands of the heart beat interval and an increase in the low-frequency ones [24]. Studies using electroencephalography (EEG) have shown that changes in cerebral electrical activity, most specifically a reduction in the power of alpha band frequencies, are related to higher stress [24,25,26,27,28]. We can therefore consider that higher HRV and alpha band values are related to lower psychological stress.

Considering that the capacity to process and select situational [29] and emotional regulation information [30] is a relevant aspect in the clinical and sports spheres, new training methods and cognitive assessment have emerged, as is the case with Neurofeedback training (NFT).

This cognitive biofeedback technique, in which an individual can learn how to modify its cerebral electrical activity, has been mostly used in therapeutic situations, with positive results in several psychiatric situations [31]. Recently, there has been an increase of its use in improving sports performance [32]. It has been proved that NFT promotes improvement of cognitive ability, reaction time and visuospatial abilities [33,34], giving individuals a base to create self-regulating strategies [35]. This, in addition to being essential for stabilization and increase in performance [36], can be improved through the use of NFT and consequently can lead to a higher performance [35,37,38,39]. These data are in conformity with previous studies that connect increased values of HRV with ANS activity and better performance [40], in addition to a reduction in stress both in athletes and in cardiovascular and chronical pain patients [36].

Bearing in mind the importance that physiological parameters such as HRV may have on athletic performance, it is important to know to what extent NFT can positively contribute to increase HRV and alpha band power. Bazanova et al. (2013) reported an increase in HRV following 10 NFT sessions aiming to increase the alpha power in a non-athlete male population [41]. Regarding HRV and NFT in sport, there are only four studies carried out, all of which compare HRV training by biofeedback with neurofeedback training [9,37,38,39], but they all assessed the effects of NFT on the athlete’s HRV.

Therefore, and considering that it is still unknown how NFT influences HRV in athletes, the current study’s aim was to determine if an α-NFT can increase HRV.

## 2. Materials and Methods

### 2.1. Subjects

Participants were randomized into two groups: (a) 3 sessions/week intervention group and (b) a 2 sessions/week intervention group [42]. A total of 30 male student-athletes aged between 18 and 34 years (mean (M) ± standard deviation (SD): 21.20 ± 2.62 for the two-session protocol vs. 22.60 ± 1.12 for the three-session protocol, *p* = 0.464) participated in the experiment. Of these 30 participants, 3 were excluded from the study due to poor-quality of the collected HRV data (1 from the 3 sessions/week group and 2 from the 2 sessions/week group). All student-athletes were provided with details about the study’s requirements before providing written informed consent to participate. Participants had to be involved in federated sports or practice regular physical activity (minimum of 30 min of at least moderate intensity 5 times a week) [43] for over 5 years [44]. The inclusion criteria were as follows: (1) all the participants had no history of psychiatric or neurological disorders; (2) no psychotropic medications or addiction drugs; (3) normal or corrected-to-normal vision; (4) minimum age of 18 years and maximum age of 35 years; and (5) to have been practicing vigorous exercise regularly at least 5 times a week regardless of skill level. All student-athletes were provided with details about the study’s requirements before providing written informed consent to participate. This study was carried out in accordance with the recommendations of local ethics guidelines and approved by the Ethics Committee of the Faculty of Human Kinetics, University of Lisbon (24/2017, approval date 26 June 2017) and in accordance with the standards for ethics in sport and exercise science research [45]. All participants gave written informed consent in accordance with the Declaration of Helsinki [46]. All data collected have been stored in a database with password protection to which only researchers related to the NFT project have access. Anonymity was guaranteed.

### 2.2. Experimental Design

The 12 NFT sessions consisted of 25 trials of 60 s each with 5 s rest in-between, during which both EEG and HRV were recorded. The total NFT session time for each subject was 300 min in both intervention groups. Naturally, the participants who performed the most frequent protocol had more condensed NFT sessions than the subjects who per-formed the less frequent protocol.

Both the 2 sessions/week and the 3 sessions/week groups performed an instruction session and a pre-test before the 12 NFT sessions. At the end of completing all NFT sessions, a post-test was performed. The instruction session consisted in 5-min NFT practice trial, for participants to understand the training feedback, with instructions being given to clarify the purpose of these procedures in the context of the study. Both pre and post-tests were carried out on the same day of the first and last training sessions, respectively. In the beginning of each session, prior to the NFT, there was also a resting baseline recording which consisted of four epochs/periods of 30 s: two with the eyes open (EO) and two with the eyes closed (EC).

The participants were asked to be as relaxed as possible and to concentrate on a specific sport task.

### 2.3. Electroencephalography (EEG)

#### 2.3.1. Data Acquisition

During the experiment, the participants sat in a room with a controlled environment—silent room with no light. The EEG signals were recorded according to the international 10–20 system (Fp1, Fp2, F3, F4, F7, F8, C3, C4, T3, T4, P3, P4, T5, T6, O1, O2, Fz, Cz, and Pz), with a sampling frequency of 256 Hz. The Cz channel was chosen for feedback since this location is at the primary motor cortex and has been associated with sensory information processing over the sensorimotor area. Furthermore, it provides a measurement of the activity in both hemispheres and in the frontal lobe [47,48].

The ground was located at the forehead and the reference was the average of left and right mastoids. The signals were amplified by a 24-channel system (Vertex 823 from Meditron Electomedicina Ltd.a, São Paulo, Brazil) and were recorded by Somnium software platform (Cognitron, São Paulo, Brazil). The signals were filtered with an analog band-pass filter from 0.1 to 70 Hz in the amplifier and a digital band-pass filter from 4 to 30 Hz. Circuit impedance was kept below 10 kΩ for all electrodes before the sessions. Subjects were asked to sit comfortably and then to remain as still as possible and to avoid excessive blinking as well as abrupt movements.

#### 2.3.2. Individual Alpha Band (IAB)

Since a large interindividual difference in the alpha band has been reported, an individual alpha band (IAB) is often used instead of a standard fixed band based on a normative population [49].

EEG recordings of EO and EC periods during the resting baseline provide data for the calculation of alpha desynchronization and synchronization, respectively, enabling to determine frequency bands for each participant through amplitude band crossings [49]. The EEG signal for the channel Cz was notch-filtered at 50 Hz and lowpass-filtered at 30 Hz. The Welch’s method was used to compute the power spectrum density for EO and EC, using an overlap of 10% and segments of 5s. The crossings between the two power spectra provide the transition frequencies to neighbouring bands: the lower frequency boundary (LB) of IAB and the upper frequency boundary (UB) of IAB. Thus, they define the IAB, which lays between the two crossings, as illustrated in [50]. The IAB information and their statistical comparisons between two NFT groups are summarized in Table 1.

### 2.4. NFT Intervention Procotol

Feedback is a determinant step for the protocol’s success. Neural activity must be fed back by some parameter(s) and presented to the participant in a simple and direct representation of their value. In this study, the feedback parameter was the relative IAB amplitude in channel Cz, which uses the amplitude from 4 to 30 Hz as a normalization factor for IAB. This is calculated as in Equation (1), where the numerator indicates the averaged amplitude in IAB, denominator indicates the averaged amplitude in 4–30 Hz, the LB is the lower frequency boundary (LB) of IAB, UB is the upper frequency boundary (UB) of IAB, and X(k) is the frequency amplitude spectrum calculated by fast Fourier transformation (FFT) with a sliding window of 2 sec that shifted every 125 ms. The frequency resolution was 0.5 Hz. Using the amplitude spectrum instead of the power spectrum prevents excessive skewing, which results from squaring the amplitude, and thus increases statistical validity [51].
(1)$$\mathrm{Relative}\;\mathrm{IAB}\;\mathrm{amplitude}=\frac{\frac{\sum_{k=LB}^{UB}X\left(k\right)}{UB-LB}}{\frac{\sum_{k=4}^{30}X\left(k\right)}{30-4}}$$

The EEG training plugin included in the Somnium software was used to provide the visual feedback and is further detailed in [50]. The visual feedback display contains two objects: the first one in the centre and a second one in the lower left corner. These two objects change their shape and position, respectively, if the feedback parameter exceeds a certain predefined threshold (goal 1) and, in that case, if the participant is able to achieve goal 1 during a predefined amount of time (goal 2).

The central object is a small white prism with a rhombus base (four-sided) that changes. If goal 1 is being achieved, the number of sides of the base increases, progressively shaping and smoothing the white prism into a bigger purple sphere. If goal 1 stops being achieved, the number of sides progressively decreases back to the initial rhombus shape, with its colour fading back to white and its size diminishing.

The second object is a cube whose position on screen is related to the period during which goal 1 kept being achieved continuously. If it happens for more than a predefined period of time (2 s), goal 2 is accomplished, and the cube moves upwards until goal 1 stops being achieved. If that happens, it will start moving downwards back to the initial position unless goal 2 is achieved again. Therefore, the participant’s task is to move the cube upwards as much as possible [50].

The feedback threshold was set to 1.0 in the first session (i.e., the quotient between the mean IAB amplitude and the EEG total average amplitude 3–40 Hz had to be larger than 1, as shown in Equation (1)) Afterwards, it was adjusted according to the percentage of time during which the feedback parameter was above the threshold in each session. If this percentage exceeded 60%, the threshold would be increased by 0.1 in the next session. In contrast, if the percentage was below 20%, the threshold would be decreased by 0.1 in the next session [52].

Although inhibiting mental self-talk seems to be one of the best strategies to achieve self-regulation of EEG activity during NFT [49,53,54,55,56], participants were instructed only to concentrate on their sport activity as much as possible. If the feedback provided on screen was positive and the goals were being achieved, that would mean their strategy was working. If not, they were encouraged to find new strategies to achieve the goals.

### 2.5. Heart Rate Variability (HRV)

#### 2.5.1. Data Acquisition

For HRV analysis the RR interval data were gathered during the training session, in accordance with the methodological considerations proposed by the European Society of Cardiology (1996) and by Billman and associates (2015) [57,58]. The cardiac cycle duration and respective RR intervals were measured using a Polar H7 (Kempele Finland) heart rate monitor strapped around the participant’s chest. The RR interval data were paired with the Elite HRV collection application, and processed and analysed using Kubios software (Kuopio, Finland).

#### 2.5.2. Root Mean Square of Successive Differences (RMSSD)

The HRV was analysed using a time-domain measure, RMSSD, which corresponds to the root mean square of the successive differences between adjacent RR intervals (RRI). Data collection started after a 30 s stabilization period, following which RR intervals were monitored throughout the duration of the task. RR interval data was recorded at a rate of 250 Hz following the equation:(2)$$\mathrm{RMSSD}=\sqrt{\frac{\sum_{i}^{N}{\left(RR_{i+1}-RR_{i}\right)}^{2}}{N-1}}$$
where (RR_i+1_ − RR_i_)^2^ corresponds to the square of the difference between RR interval time-length, N corresponds to the total number of intervals. RMSSD was chosen as the HRV variable of interest since it seems to be more related to vagal activity [57] and has shown greater reliability than other spectral variables indicators [23].

### 2.6. Statistical Analysis

The comparison of session means related to the relative amplitude of IAB and HRV were performed using Student’s t-test for independent samples and Mann-Whitney U when normality was not verified. Generalized estimating equations, followed by Bonferroni post hoc test, were used to estimate between-group and within-group effects on IAB and HRV. Data were analysed using SPSS software for Windows version 25.0 (SPSS Inc., Chicago, IL, USA). Statistical significance was established as *p* < 0.05 in all tests. Computed by G*Power software (version 3.1.9.4) for a 0.05 significance level and a 0.95 power before experiment.

## 3. Results

Differences between relative IAB amplitude and HRV means during NFT training between both protocols are presented in Table 1.

Table 2 presents the results concerning IAB and HRV at baseline and after 12 sessions, as well as the effect of interaction using per-protocol analyses for each group (time—session 1 and session 12—vs. group).

When considering the interaction time with the group, only the three-sessions protocol experienced changes in the IAB (β, 0.211; *p* = 0.018). Moreover, it was found significance between session 1 and session 12 in IAB (β, 0.237 *p* < 0.001) and in HRV (β, 15.909 *p* = 0.025).

## 4. Discussion

The aim of the study was to verify whether the NFT had direct implications for the increase in HRV. IAB was used to perform the NFT and two different groups to help clarify the extent to which this biofeedback training could impact on HRV.

Higher HRV values have been correlated with a greater information processing capacity and attention focus maintenance, suggested as important aspects in order to respond to changes in environment conditions [15]. According to Lehrer and colleagues (2000), HRV alterations due to biofeedback training in respiratory control are also associated with a higher capacity for regulating heart rate, which the authors suggested as being related to a more efficient action of baroreflexes [14]. Other interventions that led to an increase in HRV seem to be related to changes in important aspects of sports performance, such as a better technique and lower anxiety index [38].

According to data presented in Table 2, when the total training period (12 weeks) is considered, we verify that only the group that performed the most frequent training per week improved both relative IAB amplitude and HRV (RMSSD) means significantly. If an NFT is applied with a weekly frequency of three sessions, the results can clearly be achieved after 12 sessions, but the same does not happen with a lower weekly NFT frequency protocol in IAB. This lower frequency intervention will probably need more total sessions so that the effects can be observed. Even though HRV demonstrates significant changes, it is imperative to notice that had nothing to do with the group interaction. The group that performed the 3 weekly sessions had on average, a lower baseline value than the group that performed the 2 weekly sessions. However, Table 1 proves that the baseline values between groups are not significantly different, which further reinforces the effectiveness of a training in the IAB with a weekly load of three sessions.

As already mentioned, in addition to the fact that there are few studies in the sport/exercise research field that combine NFT and HRV, they only compare the effects of each training protocol without analysing how NFT can influence HRV [9,37,38,39]. Despite this, Rijken (2016) was careful to mention that although their study was not performed to understand the influence of NFT on HRV, he observed what seemed to be an association between the two variables [39]. These data are supported by a previous study that demonstrated the influence of an alpha band training (NFT) on HRV [59]. Recently, a case study was carried out on a patient with stroke and it was found that NFT had a positive effect on increasing HRV [60]. Although this is only one case and belongs to a very different from that observed in this study, the results are in agreement with our findings.

The main strength of the study and what makes it so important is that it is the first study, of our knowledge, that tries to understand the influence of an α-NFT in the increase of HRV in athletes, demonstrating that performing thee neurofeedback sessions per week can lead to improvements in HRV. The training individualization was also considered (IAB was used instead of the fixed alpha band) [49].

There were some limitations that should be considered. First, the mental strategies were not recorded. Future work should include a questionnaire or scale to better understand what strategies athletes/individuals are using during NFT and which mental strategies are helpful to enhance NFT learning of training frequency band activity [38]. The present study should therefore be considered exploratory. Additionally, no cognitive laboratory tests or stress and anxiety scales were performed to notice physiological and psychological behavioural changes, thus it is imperative to not generalize the results. Finally, it would be important to include a sham group.

## 5. Conclusions

Ultimately, the 3 sessions/week group showed effective increments in relative IAB amplitude and HRV (RMSSD) mean values in student athletes. However, no interaction between groups and HRV time were found. suggesting that NFT can indeed be an effective tool to consider in the sports domain, in order to induce changes in HRV and cognitive parameters, but has nothing to do with the weekly training frequency. Future research should replicate the three sessions/week protocol based on a pre-test and post-test associated to anxiety and stress scales and in agreement with specific sport performance to better understand how the increased alpha and consequently higher HRV contributes to a better sporting performance. Additionally, stronger conclusions could be drawn in future studies with sham control conditions.

## Figures and Tables

**Table 1 ijerph-18-12579-t001:** Descriptive differences between groups in relative IAB amplitude and HRV means.

	M ± SEM		
	Two-Session Protocol (n = 13)	Three-Session Protocol (n = 14)	*p*
IAB S1	1.56 ± 0.08	1.46 ± 0.07	0.366 ^a^
IAB S12	1.58 ± 0.11	1.69 ± 0.09	0.171 ^a^
HRV (RMSSD) S1	59.05 ± 5.50	49.68 ± 6.97	0.306 ^a^
HRV S12 (RMSSD)	65.19 ± 3.00	65.59 ± 7.09	0.960 ^a^

M, mean; SD, standard deviation; IAB, individual alpha band; HRV, heart rate variability; RMSSD, root mean square of successive differences between normal heartbeats; S, session.^a^ Differences between groups tested with Student’s t-test.

**Table 2 ijerph-18-12579-t002:** Individual alpha band and heart rate variability at session 1 and session 12: within and between protocol groups.

	**Two-Sessions Protocol (2S)**		**Three-Sessions Protocol (3S)**		**2S*3S** **β(95%CI)**
**Variables**	**Session 1**	**Session 12**	**Session 1**	**Session 12**	
IAB (n = 14)	1.56 ± 0.08	1.58 ± 0.11	1.46 ± 0.07	1.69 ± 0.09 ^b^	0.211 ^a^ (0.036; 0.386)
HRV (n = 13)	59.05 ± 5.50	65.19 ± 3.00	49.68 ± 6.97	65.59 ± 7.09 ^b^	9.768 (−7.67; 27.21)

IAB, Individual alpha band; HRV, heart rate variability. Betas are presented as unstandardized coefficients between the interaction time*group with the respective 95% confidence intervals. ^a^ Between-group changes significant at *p* < 0.05 ^b^ Within-group changes significant at *p* < 0.05.

## Data Availability

The data presented in this study are available on request from the corresponding author.

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
