# Peer review of "The Influence of an Alpha Band Neurofeedback Training in Heart Rate Variability in Athletes"

_ijerph, 2021, doi:10.3390/ijerph182312579_

Round 1
Reviewer 1 Report
This manuscript presents an interesting research question and approach, but I have some major reservations about the statistical approach.
First, a few issues with the Introduction. A major concern is that there is no background literature on why the authors expect an alpha band effect of NFT. What is the theoretical background that links alpha band to (high-frequency) HRV? Why would the authors expect NFT of the alpha band to lead to increases in (high-freq) HRV? A more minor concern is the invocation of "sympathovagal function" in the 2nd paragraph. This should be heavily caveated or just removed, as the notion of sympathovagal balance has been debunked. Heathers and colleagues have shown that low-frequency HRV does not serve as index of sympathetic nervous system activity, and further, that taking low-to-high-frequency ratio does not approximate some ratio of sympathetic to parasympathetic influence (Heathers, 2012; Heathers, 2014; Quintana & Heathers, 2014). There is plenty of literature showing that high-frequency heart rate variability is related to cognitive performance and emotion regulation, implicating a PNS association, and it is critical for the authors to distinguish this from SNS ("sympatho-") function.
In the Methods section: "mean IAB amplitude" is written with no prior explanation of what IAB is. Information on calculating IAB appears at the very end of the Methods section, and I think the text would flow better if IAB was either included in the EEG section, or in a section more immediately after it. The current organization is EEG (2.2.1) to HRV (2.2.2) to Experimental Design (2.3) then back to the EEG Measurement (2.4). This gave me much confusion as a reader. Also, the authors should specify how they determined frequency bands individually "through amplitude band crossings", as setting the lower and upper bounds of alpha hugely influences the resulting measure (and as individuals vary in bandwidths that demonstrate alpha waves).
As for my major reservations, the statistical approach is quite liberal. Running separate paired samples t-tests to test S1, S12, and S12-S1 separately for each group and separately for IAB and HRV, could all be better captured by a repeated measures ANOVA with a two-level factor of Time (S1 and S12) and a two-level between-subjects Group factor (the two NFT protocols). It's not clear to me that the difference between IAB (S12-S1) at p=.03 would be supported by a significant Group x Time interaction. The benefit of running a repeated measures ANOVA would also be to better clarify any main effects between the NFT protocols (main effect of Group) and clarifying effects of repeated testing (main effect of Time). One reason this is absolutely critical is that it looks like the 3 session group has significantly lower IAB and HRV at the 1st measurement point than the 2 session group. Thus, the 3 session group may have more "room" to improve.
The pattern of results reported here is very compelling, but without a more robust statistical approach, I worry that the conclusions drawn here are inappropriate. It is imperative that the authors demonstrate that the two groups do not differ at time 1, and if they do, that any differences in gains over time between groups are statistically meaningful while controlling for time 1 values.
Author Response
Reviewer_1:
Dear Reviewer,
We want to thank you for your suggestions and questions. It's really rewarding to see that the issues raised will help to greatly improve our work!
Point 1.
1.1. First, a few issues with the Introduction. A major concern is that there is no background literature on why the authors expect an alpha band effect of NFT. What is the theoretical background that links alpha band to (high-frequency) HRV? Why would the authors expect NFT of the alpha band to lead to increases in (high-freq) HRV? 1.2. A more minor concern is the invocation of "sympathovagal function" in the 2nd paragraph. This should be heavily caveated or just removed, as the notion of sympathovagal balance has been debunked. Heathers and colleagues have shown that low-frequency HRV does not serve as index of sympathetic nervous system activity, and further, that taking low-to-high-frequency ratio does not approximate some ratio of sympathetic to parasympathetic influence (Heathers, 2012; Heathers, 2014; Quintana & Heathers, 2014). 1.3 There is plenty of literature showing that high-frequency heart rate variability is related to cognitive performance and emotion regulation, implicating a PNS association, and it is critical for the authors to distinguish this from SNS ("sympatho-") function.
Response 1.
1.1. We greatly appreciate your careful reading of the introduction and recognize that we were missing a background directly relating alpha band and HRV. We included a whole paragraph with recent literature to support our thesis [1-5]. You can find the info in lines [62-67]. Regarding alpha NFT and HRV, to our knowledge, only Bazanova et al. (2013) investigated HRV changes following NFT aiming to increase alpha power, and they reported an increase in HRV with NFT [6]. Nonetheless, their work did not study athletes, so we consider that it does not overlap significantly with ours, although it represents a major motivation performing our work as we also expected an increase of HRV following NFT.
1.2. Thank you for your detailed explanation! Regarding the fact that low-to-high-frequency ratio does not approximate some ratio of sympathetic to parasympathetic influence, we have reviewed the recommended literature and have decided to remove our reference to "sympathovagal regulation" altogether. We decided not to dig into the physiological underpinnings of HRV as these were out of the scope of the present article.
1.3. In our sincere opinion we've presented enough pertinent information concerning the importance of HRV for cognitive and physical performance- a higher variability in the rhythmic oscillation of heart rate seems to be correlated with better health and performance outcomes, and how does NFT influence HRV related variables.
Point 2.
2.1. In the Methods section: "mean IAB amplitude" is written with no prior explanation of what IAB is. Information on calculating IAB appears at the very end of the Methods section, and I think the text would flow better if IAB was either included in the EEG section, or in a section more immediately after it. The current organization is EEG (2.2.1) to HRV (2.2.2) to Experimental Design (2.3) then back to the EEG Measurement (2.4). This gave me much confusion as a reader. 2.2. Also, the authors should specify how they determined frequency bands individually "through amplitude band crossings", as setting the lower and upper bounds of alpha hugely influences the resulting measure (and as individuals vary in bandwidths that demonstrate alpha waves).
Response 2.
2.1. Thank you for your kind suggestions, we agree with your comments regarding the IAB definition and the organization of the methods sections.
Furthermore, we have reorganized the Methods section to provide first a description of the subjects (2.1 Subjects) and design (2.2 Experimental Design), then all the EEG-related information (2.3. Electroencephalography (EEG); 2.3.1. Data acquisition: 2.3.2 Individual Alpha Band (IAB)), followed by the NFT protocol (NFT intervention protocol), HRV (2.5 Heart Rate Variability (HRV); 2.5.1 Data Acquisition; 2.5.2 Root Mean Square of Successive Differences (RMSSD)) and finally the statistical analysis (2.6 Statistical Analysis).
2.2. In the revised version, we have now included a detailed description of how the IAB is computed, clarifying how we use the amplitude crossings. This corresponds to the second part of the EEG subsection, following the data acquisition, and is entitled “Individual Alpha Band (IAB)”.
Point 3.
3.1. As for my major reservations, the statistical approach is quite liberal. Running separate paired samples t-tests to test S1, S12, and S12-S1 separately for each group and separately for IAB and HRV, could all be better captured by a repeated measures ANOVA with a two-level factor of Time (S1 and S12) and a two-level between-subjects Group factor (the two NFT protocols). It's not clear to me that the difference between IAB (S12-S1) at p=.03 would be supported by a significant Group x Time interaction. The benefit of running a repeated measures ANOVA would also be to better clarify any main effects between the NFT protocols (main effect of Group) and clarifying effects of repeated testing (main effect of Time). 3.2. One reason this is absolutely critical is that it looks like the 3 session group has significantly lower IAB and HRV at the 1st measurement point than the 2 session group. Thus, the 3 session group may have more "room" to improve.
Response 3.
3.1. Since we are talking about results, it is imperative to explain that we did some big changes. First, we added Descriptive in the Table 1. caption to help readers to realise that we are comparing between groups without seeking to understand the influence of NFT in HRV (it led to some confusion in reviewer’s 2 interpretation). We also excluded the differences between IAB and differences between HRV to avoid more misunderstandings.
Now, we deleted Figure 1. and replaced it to Table 2. because we understand that the slope is not the right measure to consider since we are doing a generalized linear model and comparing only the moment 1 and moment 12 and not all the variation during the session.
Re-doing the statistic was our most important improvement. However, we did not perform an ANOVA because some variables were not normal. Instead, we used generalized estimating equations. You were right, even though we had improvements in IAB and HRV, we only found interaction between group vs time in IAB.
Table 2. Individual alpha band and heart rate variability at session 1 and session 12: within and between protocol groups.
|
|
Two-sessions protocol (2S) |
Three-sessions protocol (3S) |
2S*3S β(95%CI) |
||
|
Variables |
Session 1 |
Session 12 |
Session 1 |
Session 12 |
|
|
IAB (n = 14) |
1.56 ± .08 |
1.58 ± .11 |
1.46 ± .07 |
1.69 ± .09b |
.211a (.036; .386) |
|
HRV (n = 13) |
59.05 ± 5.50 |
65.19 ± 3.00 |
49.68 ± 6.97 |
65.59 ± 7.09b |
9.768 (-7.67; 27.21) |
|
IAB, Individual alpha band; HRV, heart rate variability Betas are presented as unstandardized coefficients between the interaction time*group with the respective 95% confidence intervals. a Between-group changes significant at p < 0.05 b Within-group changes significant at p < 0.05 |
|||||
3.2. The statistic mentioned above helped us in that domain, nevertheless, Table 1. already showed that the baseline in both groups were not significantly different.
References
- Al-Shargie, F., T.B. Tang, N. Badruddin, and M. Kiguchi. Simultaneous measurement of EEG-fNIRS in classifying and localizing brain activation to mental stress. in 2015 IEEE International Conference on Signal and Image Processing Applications (ICSIPA). 2015. IEEE.
- Singh, Y. and R. Sharma, Individual Alpha Frequency (IAF) Based Quantitative EEG Correlates of Psychological Stress. Indian J Physiol Pharmacol, 2015. 59(4): p. 414-21.
- Alonso, J.F., S. Romero, M.R. Ballester, R.M. Antonijoan, and M.A. Mananas, Stress assessment based on EEG univariate features and functional connectivity measures. Physiol Meas, 2015. 36(7): p. 1351-65 DOI: https://10.1088/0967-3334/36/7/1351.
- Zhu, L., X. Tian, X. Xu, and L. Shu. Design and Evaluation of the Mental Relaxation VR Scenes Using Forehead EEG Features. in 2019 IEEE MTT-S International Microwave Biomedical Conference (IMBioC). 2019.
- Katmah, R., F. Al-Shargie, U. Tariq, F. Babiloni, F. Al-Mughairbi, and H. Al-Nashash, A Review on Mental Stress Assessment Methods Using EEG Signals. Sensors (Basel), 2021. 21(15): p. 5043 DOI: 10.3390/s21155043.
- Bazanova, O., N. Balioz, K. Muravleva, and M. Skoraya, Effect of voluntary EEG α power increase training on heart rate variability. Human Physiology, 2013. 39(1): p. 86-97 DOI: https://10.1134/S0362119712060035.

Reviewer 2 Report
I strongly agree that NFT and HRV biofeedback training have very important aspects in athletic performance, and brain and heart activity are connected. More extensive research are required in order to examine the dynamic relationships in central nerve system and automatic nerve system regarding athletic performance and arousal regulation. Even though this study introduced an important topic in sport and exercise psychology, there are some regretful parts in methodology and results.
In terms of research design, the participants were divided into two group, such as two sessions per a week and three sessions per a week. I don’t think it is an appropriate comparison. How can they be independently comparable? It directs me to the result in Table 1. There should have been a control group to be comparable. The results in table 1, indicated that there was no effect of NFT on HRV changes both in two- and three- session groups.
In methodology, I would like to see the NFT intervention protocol under a different heading. It is not a measurement. Also, HRV measurement (RMSSD) protocol and formula are missing.
In results, descriptive statistics should be shown for readers. In regression analysis, p-value corresponding to each slope should be reported. As far as I understand, HRV was increased as sessions went, right? Then, two regression models are meaningless. And the comparison between the two slopes, as saying that three-sessions was more effective, is not valid.
In overall conclusion, I think more NFT is effective on HRV changes, however it is not matter of two- or three- sessions per a week.
Author Response
Reviewer_2
Point 1.
I strongly agree that NFT and HRV biofeedback training have very important aspects in athletic performance, and brain and heart activity are connected. More extensive research are required in order to examine the dynamic relationships in central nerve system and automatic nerve system regarding athletic performance and arousal regulation. Even though this study introduced an important topic in sport and exercise psychology, there are some regretful parts in methodology and results.
Response 1.
We appreciate your time and suggestions, as they have provided valuable help towards improving our work.
Point 2.
In terms of research design, the participants were divided into two group, such as two sessions per a week and three sessions per a week. 2.1. I don’t think it is an appropriate comparison. How can they be independently comparable? It directs me to the result in Table 1. There should have been a control group to be comparable. 2.2. The results in table 1, indicated that there was no effect of NFT on HRV changes both in two- and three- session groups.
Response 2.
2.1. I appreciate your honesty in sharing your concerns. We used the 3 sessions vs 2 sessions per week group, because a study published this year by our group revealed that the alpha band varied significantly. Therefore, we wanted to know if the weekly frequency would also have the same result in HRV. But again, we really understand your concerns and even reported it as limitations of our study (lines 423 – 424).
2.2. I believe we weren't clear enough with Table 1. Table 1 only compares differences between groups, independently. No statistical analysis is performed to find out whether the IAB has an influence on HRV. We just compared the time 1 values with the time 1 values and the time 12 values with the time 12 values for the variables IAB and HRV, independently. We removed the difference between the moments, so it won’t cause possible misunderstandings.
Point 3.
3.1. In methodology, I would like to see the NFT intervention protocol under a different heading. It is not a measurement. 3.2. Also, HRV measurement (RMSSD) protocol and formula are missing.
Response 3.
3.1. Thank you for your suggestion. We changed the heading. We've also tried to be clearer when it comes to the relevant HRV metrics and the methodology used for data collection and analysis, considering your feedback.
3.2. Thank you. We have added these details in the subsection “2.5.2 Root Mean Square of Successive Differences (RMSSD)”.
Point 4.
4.1. In results, descriptive statistics should be shown for readers. 4.2. In regression analysis, p-value corresponding to each slope should be reported. As far as I understand, HRV was increased as sessions went, right? Then, two regression models are meaningless. And the comparison between the two slopes, as saying that three-sessions was more effective, is not valid.
Response 4.
4.1. As mentioned above, probably because we weren’t clear enough, our table 1 was designed to be descriptive. We can add the age (only variable left), but we already mentioned it in the text (subjects’ sub-section).
4.2. Since we had to re-do our statistic (suggested by reviewer 1), we excluded Figure 1 because your point regarding the slopes is valid, and it is indeed meaningless. We performed a generalized estimating equation since one of our variables was not normal (instead of ANOVA two factors).
Table 2. Individual alpha band and heart rate variability at session 1 and session 12: within and between protocol groups.
|
|
Two-sessions protocol (2S) |
Three-sessions protocol (3S) |
2S*3S β(95%CI) |
||
|
Variables |
Session 1 |
Session 12 |
Session 1 |
Session 12 |
|
|
IAB (n = 14) |
1.56 ± .08 |
1.58 ± .11 |
1.46 ± .07 |
1.69 ± .09b |
.211a (.036; .386) |
|
HRV (n = 13) |
59.05 ± 5.50 |
65.19 ± 3.00 |
49.68 ± 6.97 |
65.59 ± 7.09b |
9.768 (-7.67; 27.21) |
|
IAB, Individual alpha band; HRV, heart rate variability Betas are presented as unstandardized coefficients between the interaction time*group with the respective 95% confidence intervals. a Between-group changes significant at p < 0.05 b Within-group changes significant at p < 0.05 |
|||||
Point 5.
In overall conclusion, I think more NFT is effective on HRV changes, however it is not matter of two- or three- sessions per a week.
Response 5.
After adding the generalized estimating equation, we concluded you were correct. Thank you for your help.

Round 2
Reviewer 2 Report
The manuscript has been improved in method and result. It is now clear enough to be published in this journal